# Assessing anti-rabies vaccine response in humans: A rapid and high-throughput adaptable, pseudovirus-based neutralization assay as an alternative to rapid fluorescent focus inhibition test (RFFIT)

**Santhik S. Lupitha[1], Geetu Rose Varghese[1], Lekshmi J. Das[1], Priya Prabhakaran[2], Ashwini M. Ananda[3], Reeta S. Mani[3]\*, Easwaran Sreekumar [1,4]\***

**1** Molecular Bioassay Laboratory, Institute of Advanced Virology (IAV), Thiruvananthapuram, Kerala, India, **2** Department of Virus Applications, Institute of Advanced Virology (IAV), Thiruvananthapuram, Kerala, India, **3** Department of Neurovirology, National Institute of Mental Health and Neurosciences (NIMHANS), Bangalore, Karnataka, India, **4** Molecular Virology Laboratory, BRIC-Rajiv Gandhi Centre for Biotechnology (BRIC-RGCB), Thiruvananthapuram, Kerala, India

\* drreeta@gmail.com (RSM); esreekumar@iav.res.in, esreekumar@rgcb.res.in (ES)

## Abstract

### Background

Rabies is a fatal, but vaccine-preventable disease. The conventional 'gold standard' recommended for estimating anti-rabies vaccine immune response is Rapid Fluorescent Focus Inhibition Test (RFFIT). Here, we describe a pseudovirus-based neutralization assay (PVNT) as an alternative method to detect the neutralizing antibody response.

### Methodology/Principal Findings

We used a recombinant Vesicular Stomatitis Virus (rVSV)-based pseudotyping system in HEK293 cells transfected with rabies G glycoprotein to produce high titre pseudoviruses. These were confirmed by neutralization with anti-G monoclonal antibodies and a reference serum, as well as immunofluorescence-based detection of pseudovirus infection. A secreted alkaline phosphatase (SEAP) reporter was included to make the assay high-throughput adaptable with readouts possible using a conventional ELISA reader. A PVNT was developed that had a dynamic range of $OD_{405}$ from 0.10 to 2.00, and was used to test a set of RFFIT-validated 71 human serum samples. Results of PVNT and RFFIT showed significant concordance with a Pearson's correlation value of 0.8555. In samples with border-line RFFIT positive values (a RFFIT titre in the range of 0.5-1.8 IU/mL), concordance was low, with the PVNT results tending to be negative. PVNT assay had a specificity of 100% when compared to RFFIT. The assay also had 100% positive predicative value indicating its robustness for use in the evaluation for the presence of neutralizing antibodies against a near 100% fatal disease such as rabies.

**Data availability statement:** All data pertaining to the study are made available within the manuscript as well as in the Supporting Information.

**Funding:** This work was supported by the Department of Biotechnology (DBT), Ministry of Science & Technology, Government of India (DBT-SAHAJ program (BT/INF/22/SP53419/2024) to ES) and by the Government of Kerala (Intramural funding of IAV Flagship program to ES). The funders had no role in study design, data collection and analysis, decision to publish, or preparation of the manuscript.

**Competing interests:** The authors have declared that no competing interests exist.

## Conclusions/Significance

Our results show that PVNT is a reproducible, sensitive and specific assay with an objective and documentable readout. The assay is faster with a turnaround time of less than 24h and can be automated for large scale screening. Importantly, it can be performed without the restrictions in handling live rabies virus. PVNT provides a reliable alternative to RFFIT for monitoring vaccine efficacy and immunity status, making it a valuable tool in rabies control programs.

## Author summary

Rabies causes approximately 59000 deaths every year world-wide. It is also among one of the neglected tropical diseases aimed to achieve 'Zero by 30' as per the UN Sustainable Development Goals (SDG) 2030. Rabies is a preventable disease both by pre- and post-exposure vaccination. One of the key requirements of successful vaccination is the induction of a robust neutralizing antibody response against the glycoprotein G, a key viral protein that facilitates the virus infection. The level of antibodies against the protein is considered as a surrogate marker of protection against rabies. The World Health Organisation (WHO) recommends a minimum serum level of 0.5 IU/mL of neutralizing antibodies to consider a person to have adequately seroconverted following vaccination. In high risk groups such as animal handlers, forest officials, pet owners and veterinarians, ensuring protective levels of anti-rabies antibodies post-prophylactic vaccination is of paramount importance. Also, population level screening of protective antibody response as part of rabies control programs in high-risk communities as well as in persons undergoing post-exposure vaccination may be essential. However, the gold-standard assays currently recommended by WHO such as RFFIT are cumbersome to perform due to multiple reasons. The present study attempts to address these issues and reports a novel assay that can overcome these difficulties. The effort is to expand public health accessibility to anti-rabies antibody testing to ensure efficacy of rabies immunization, thereby contributing to global rabies control and elimination.

## Introduction

Rabies virus, a negative-stranded RNA virus of the *Rhabdoviridae* family, is a classical zoonotic virus and an one health pathogen [1,2]. It continues to pose a significant threat in many parts of the world, especially in Asia and Africa [3,4], during a time when newly emerging viruses gain major global public health concern. The disease rabies is near 100% fatal and is transmitted primarily by animal bites. Prompt initiation of post-exposure prophylaxis including wound management, rabies vaccination, and immunoglobin administration in high-risk exposures is extremely critical to prevent the development of the disease and mortality [4,5].

Rabies virus has a 12 kb single-stranded RNA genome that encodes five proteins viz. nucleoprotein (N), phosphoprotein (P), matrix protein (M), glycoprotein (G), and an RNA-directed RNA polymerase (L), among which the protein G is the major surface glycoprotein that binds to the cellular receptor and mediate virus entry [6]. It elicits the neutralization antibody response to impart protection in pre- and post-exposure immunization [7]. Adequate pre-existing levels of anti-G neutralizing antibody prevent the entry of the virus into neurons and the central nervous system. The WHO recommends an anti-G neutralizing antibody

level of 0.5 IU/mL in the serum of vaccinated humans and animals as evidence of adequate seroconversion following vaccination [4,5]. Ensuring protective levels of virus-neutralization antibodies following vaccination in humans and animals is important to achieve rabies control as a public health measure.

Multiple approaches are employed to estimate the anti-G antibody levels in response to anti-rabies vaccination [8]. One of the earlier assays used for the purpose was the mouse neutralization test (MNT) developed in 1935 [9]. Other animal models were subsequently developed to evaluate the efficacy of new rabies vaccine candidates [10,11]. RFFIT (Rapid fluorescent focus inhibition test) and FAVN (Fluorescent antibody virus neutralization) tests are considered as 'gold standards' which are used to check the neutralization potency of anti-rabies immunoglobulins and immunogenicity of rabies vaccines [12,13]. Enzyme-linked immune-sorbent assay (ELISA) is used as a screening test in some laboratories as an alternative to RFFIT [14,15]. While inactivated rabies virus or recombinant antigens can be used as a coating antigen in ELISA and the assay will help to identify the presence of binding antibodies, it cannot be used to assess the level of the neutralizing antibody response [16]. In order to overcome this limitation of conventional ELISA-based systems, competitive and blocking ELISAs have been proposed, however, they are yet to be used in large scale analysis of real-world samples [8].

Pseudoviruses are successfully employed to study highly pathogenic viruses such as Ebola, Hepatitis C, Influenza, and Middle Eastern respiratory syndrome (MERS) viruses in BSL-2 facilities [17–20]. In these systems, the pseudovirion core belongs to a heterologous RNA virus such as human immunodeficiency virus (HIV) or vesicular stomatitis virus (VSV), and the envelope protein is of the virus of interest, which mediates the infection process in the target cells. Even though HIV-based pseudovirions of rabies virus have been generated and used in neutralization assays, the low virus titre produced in the system was a limitation [21]. Later, high titre, lentivirus-based pseudovirion systems were developed for rabies; however, the lack of high-throughput adaptability of the system prevented its widespread application [22]. Recombinant VSV (rVSV) based pseudovirus systems have been designed with removing the encoding virus surface glycoprotein G gene (ΔG). Once transduced in cells expressing a heterologous virus surface glycoprotein, it can incorporate this protein and produce infectious pseudovirus. Subsequently, in the target cells used for infection assay, these pseudoviruses can perform one round of infection and expression of the encoded genes, but cannot produce any infectious progeny virions as there are no viral surface glycoproteins being expressed. This improves the safety levels of such systems [23].

In the present study, we used this system to develop a high-throughput adaptable neutralization assay for estimating rabies virus anti-G glycoprotein antibodies. The rVSV used in our study genetically encodes Green Fluorescent protein (GFP) or Secreted Alkaline Phosphatase (SEAP) that is expressed in the infected target cells; which, in turn, support qualitative and quantitative estimation of the infection levels. We further evaluated RFFIT positive and negative human serum samples for their ability to neutralize the infectivity of the generated rabies pseudovirions.

## Materials and methods

### Ethics statement

Human serum samples were originally tested by RFFIT for anti-rabies antibody levels as per published protocols [26] at the Neurovirology Department, National Institute of Mental Health and Neurosciences (NIMHANS), Bangalore against WHO Reference serum (30IU/mL) obtained from National Institute of Biological Standards and Control (NIBSC), UK. The

left-over samples stored after RFFIT were used in the study for validation of the R-PV-SEAP pseudovirion neutralization Test (PVNT); and were completely anonymised. Hence, individual consents were not obtained for using the samples for the study. Approvals from the Institutional Human Ethics Committees of NIMHANS, Bangalore and Institute of Advanced Virology (IAV), Thiruvananthapuram (Approval No. IHEC/IAV/2023/05) were obtained for the study.

The samples included serum obtained from human subjects who received rabies vaccination (n = 61) and negative serum samples obtained from human subjects who had not received anti-rabies vaccination at any time in the past, (n = 10). Serum samples were heat-inactivated at 56 °C for 30 minutes and stored at −20 °C.

## Reagents for pseudovirus generation & Plasmids

Recombinant VSV pseudoviruses (rVSV) encoding GFP and SEAP reporters (G*VSV -ΔG-GFP and G*VSV-ΔG-SEAP, respectively) for the generation of pseudotyped rabies viruses were obtained from Kerafast Inc., Boston, MA. Plasmid encoding full-length rabies G glycoprotein (pHMC122) was a kind gift from Dr. Erica Ollman Saphire, La Jolla Institute of Immunology, USA.

## Cell lines and antibodies

Cell lines, HEK293 (ATCC, CRL1573), HEK293T (ATCC, CRL3216), and Vero cells (ATCC, CCL81) originally obtained from American Type Culture Collection (ATCC) were propagated in 10% Dulbecco's Modified Eagle's Medium (DMEM) at 37 °C under 5% $CO_2$. The media was supplemented with 10% FBS (#A525671; Invitrogen) and 1X antibiotic cocktail (#A5955; Sigma-Aldrich). Anti-rabies human monoclonal antibody, 17C7 (Rabishield-100; 40IU/ml) was purchased from Serum Institute of India Pvt Ltd; and antibody against Flavivirus group specific antigen (4G2) used as the negative control antibody was obtained from MyBiosource, USA (Cat.No.MBS488169). WHO Reference serum was obtained from the National Institute of Biological Standards and Control (NIBSC), UK.

## Transient transfection of HEK293 cells and immunostaining of rabies G protein

HEK293 cells were transiently transfected with pHMC122 plasmid expressing the full-length rabies G protein. The transfection was performed when the confluency of the cells was about 60% – 70% in an 8-well chambered cover glass (#155411; Nunc Lab-tek). 1μg of the plasmid and PEI (2.5 μg/μl; #408727 Sigma-Aldrich) were mixed in Opti-MEM with a ratio of 1:3, which was then incubated for 30 min at room temperature and was incubated with HEK293 cells at 37 °C under 5% $CO_2$. After 6h of incubation, media change was given using 10% DMEM and cells were incubated for another 12 h and fixed using ice-cold Methanol-Acetone (1:1) for 10 min at 4 °C. The cells were blocked using 3% BSA in PBST for 1hr and incubated overnight with rabies G monoclonal antibody (17C7) (1:2000) conjugated with Alexa 488 using Alexafluor 488 microscale protein labelling kit (#A30006) as per manufacturer's directions. The cells were stained with DAPI (1μg/ml; #9542 Sigma-Aldrich) for 10min at room temperature and imaged under Leica Stellaris 5 confocal microscope equipped with 40X objective (3X Zoom).

## Production of rabies pseudoviruses R-PV-GFP and R-PV-SEAP

Rabies pseudoviruses were produced from HEK293 cells transiently transfected with pHMC122 plasmid as described above. 18h post-transfection, the medium was removed

and the cells were infected with either (G*ΔG-GFP) or (G*ΔG-SEAP) rVSV pseudoviruses at an MOI of 3–5 for 2h. The medium was replaced with DMEM containing 10%FBS.The following day, the supernatant containing rabies pseudovirions R-PV-GFP and R-PV-SEAP, respectively, were harvested, clarified at 3000 rpm for 10 min, and filtered using a syringe filter (0.45μm pore size) and stored at −80 °C in aliquots until use.

In order to identify the suitable target cells for neutralization assays, the R-PV-GFP virus stocks were used to infect HEK293T and Vero cells and the GFP fluorescence was documented after 12h of infection.

## Titration of R-PV-SEAP pseudovirus stocks

Prior to use in the neutralization assays, 50% tissue culture infectious dose ($TCID_{50}$) of each batch of R-PV-SEAP pseudovirions produced was determined. To avoid inconsistency that may result from repeated freeze-thaw, virus samples were stored as single-use aliquots. The $TCID_{50}$ was calculated using Reed-Muench method [24]. Serial two-fold dilutions of the stock virus in 2% FBS containing DMEM were made and used to infect HEK293T cells in a 96-well culture plate, octuplicate wells for each dilution. A media change was given after 2h of pseudo-virion incubation and the infected cells were incubated overnight at 37 °C. The following day, the supernatant was harvested, heat inactivated at 65 °C and were analysed for SEAP activity by mixing 30μl of supernatant with 100 μl of pNPP substrate (Sigma, Cat: P7998). The reaction was stopped using 50μl 3N NaOH solution after 10 min and absorbance at 405nm was measured using Promega GloMax Discover Microplate Reader. As described in previous studies to define a positivity cut-off [25], samples with an $OD_{405}$ value greater than 3SD (standard deviation) of the $OD_{405}$ value of cell controls were considered positive for pseudovirus infection; and used in calculation of the $TCID_{50}$.

## Immunostaining of R-PV-SEAP pseudovirions post-infection in target cells

HEK 293 T cells seeded on coverslips were infected with Rabies G*(ΔG- SEAP) rVSV or VSV G*(ΔG- SEAP) rVSV pseudovirion (2500 $TCID_{50}$) at room temperature for 30 minutes followed by fixation in acetone-methanol for 5minutes. The cells were blocked in 5% BSA in PBST for 1 hour followed by incubation with primary antibody (17C7- 1:1000) overnight at 4 °C with shaking. Further, the cells were incubated antihuman Alexa Fluor 488 secondary antibody for 1hour. DAPI was used as the nuclear stain. The cells were stained with DAPI (1μg/ml) for 10min at room temperature and imaged under Leica stellaris5 confocal microscope.

## R-PV-GFP and R-PV-SEAP pseudovirion- based PVNT

For the PVNT assay, test human serum samples, the positive control monoclonal antibody 17C7 (2IU/mL) (titrated against the WHO reference serum) and the negative control antibody 4G2 each were diluted two-fold in 2% FBS containing DMEM. R-PV-GFP or R-PV-SEAP pseudovirions were incubated with serially diluted antibodies or serum for 1hr at 37 °C. R-PV-GFP pseudovirions containing transduced producer HEK293 cell culture supernatants were used undiluted for neutralization assays. R-PV-SEAP pseudovirions were used at a concentration of 100$TCID_{50}$. HEK293T target cells ($2 \times 10^4$ cells/ well) were seeded in the 96 well plates; and after 24h incubation, the medium was removed and the mono-layers were infected with the pseudovirion-antibody mixture for 2h. A media change was given using 10% DMEM and cells were incubated overnight. For R-PV-GFP-infected cells 10X magnification images were captured using a Zeiss AXIO Vert.A1 inverted fluorescence microscope equipped with AXIOCAM 305 colour camera using FITC excitation-emission filter settings.

The supernatant containing secreted alkaline phosphatase (SEAP) enzyme from R-PV-SEAP-infected cells was harvested, endogenous phosphatase activity was inactivated by heating at 65 °C for 10 min and SEAP activity was measured as described above. Percentage neutralization was calculated using the formula:

$$\left[1-\frac{\left(\text{OD}_{\text{Sample}}-\text{OD}_{\text{Cell control}}\right)}{\left(\text{OD}_{\text{Virus control}}-\text{OD}_{\text{Cell control}}\right)}\right]\times 100$$

For identifying the 50% endpoint dilution for calculation of the PVNT titre, the 71 test serum samples and the WHO reference serum were diluted 2-fold from 1:64–1: 8192, as followed for the RFFIT titre evaluation [26] and subjected to PVNT; and 50% end point was identified Reed-Muench method [24]. The PVNT Titre for each of the serum sample was as assigned using the equation:

$$\text{PVNT Titre} =\left(\frac{\text{Reciprocal of 50\% endpoint dilution of test serum}}{\text{Reciprocal of 50\% endpoint dilution of Reference serum}}\right)\times$$
$$\text{Unitage of reference serum}$$

## Analysis of sensitivity, specificity, correlation analysis and receiver operator characteristics (ROC) of the PVNT assay

Results of PVNT and RFFIT assays of the human serum samples were compared to calculate the number of true positives (TP), true negatives (TN), false positives (FP) and false negatives (FN). Further, specificity [i.e.,TN/(FP+TN)], sensitivity [i.e., TP/(TP+FN)], positive predictive value (PPV) [i.e., TP/(TP+FP)], and negative predictive value (NPV) [i.e., TN/(TN+FN)] of the PVNT assay was calculated.

## Statistical analysis

The Pearson's correlation analysis of the PVNT and RFFIT titres; and the receiver operator characteristics (ROC) of the assay as a relation of Sensitivity vs (1-Specificty) were plotted using GraphPad Prism version 6.

## Results

### Generation of pseudovirion R-PV-GFP and evaluation of infectivity

GFP expressing rabies pseudovirions (R-PV-GFP) were generated as per the scheme shown in Fig 1A. A high level expression of rabies G protein on the membrane of the producer packaging cell line HEK293 is required for efficient pseudotyping with rVSV. This was confirmed by immunostaining with anti-rabies G 17C7 antibody conjugated with Alexa Fluor 488. As shown in Fig 1B, in confocal microscopy, a high-level membrane expression of the G protein was clearly visible on the pHMC122 plasmid-transfected cells 12h post-transfection.

The infectivity of R-PV-GFP was tested by observing the GFP signal upon infecting HEK293T and Vero cells as the target cells. These cells were infected with the clarified supernatant collected 12h after rVSV transduction from the pHMC122 or mock plasmid transfected HEK293 producer cells. As shown in Fig 1C, the supernatant from pHMC122 transfected cells contained infectious pseudovirions of R-PV-GFP indicated by the significantly large numbers of fluorescence foci in the target cells. As expected, there were a very small number of fluorescent spots in the cells infected with supernatant from mock-plasmid transfected cells. These indicate the carry over presence of VSVG expressing virus in the

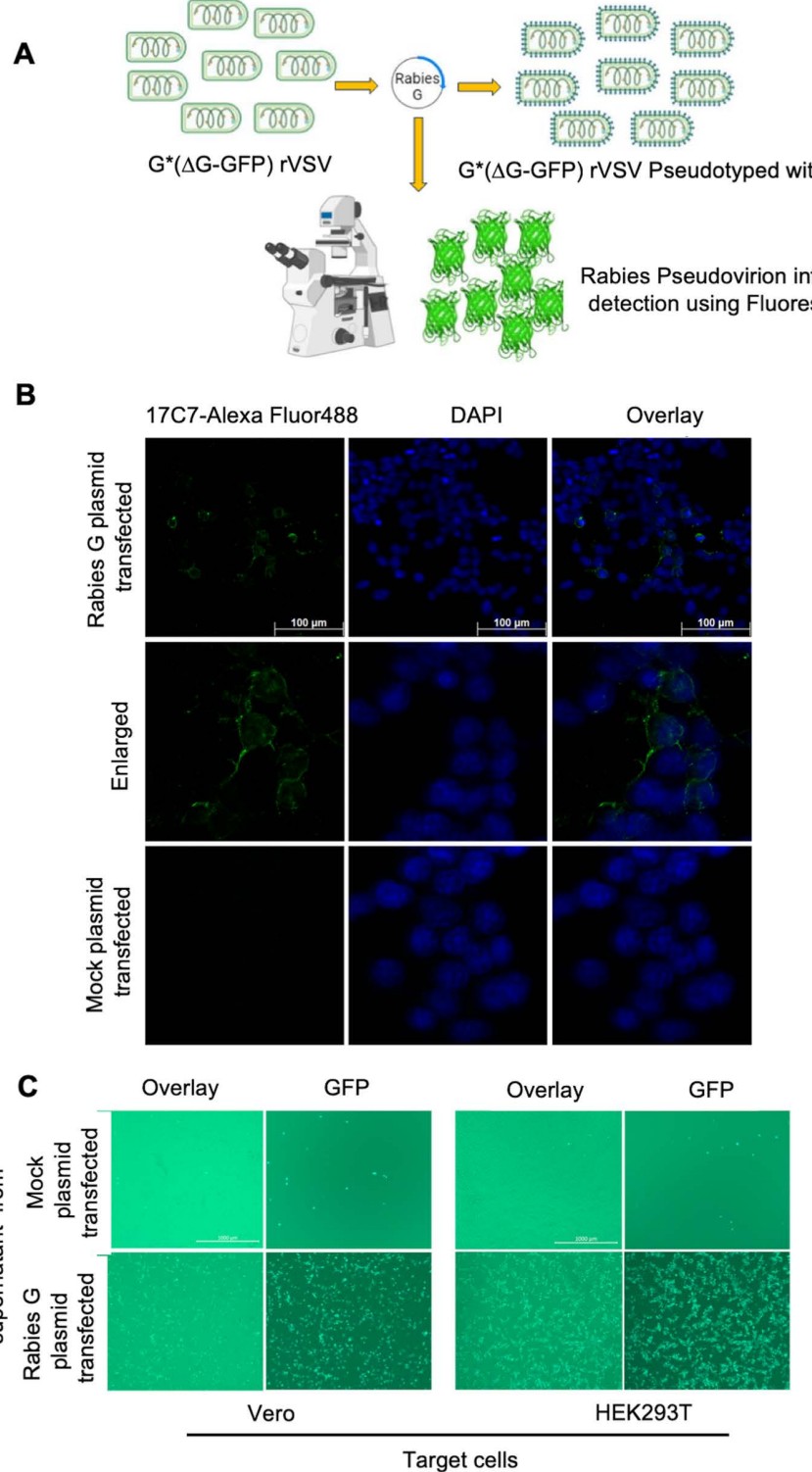

**Fig 1. Generation and Characterization of infectivity of R-PV-GFP. A.** Scheme showing generation of Rabies pseudovirus expressing GFP reporter (R-PV-GFP) using rVSV (G*ΔG-GFP) and full-length Rabies virus G protein expressing plasmid pHMC122 (Generated using Biorender; https://www.biorender.com/). **B.** Immunofluorescence analysis of Rabies virus G protein expression in HEK293 producer cells used for pseudovirion generation. Cells transfected with full-length G protein expressing plasmid were immunostained with 17C7 monoclonal antibody

conjugated with Alexafluor 488, 24h post-transfection. Nuclei were counter-stained with DAPI and cells were imaged under a Leica Stellaris 5 confocal microscope equipped with 40x objective (3x Zoom). **C.** Supernatants (100 μl) containing pseudovirions generated from HEK293 producer cells transfected either with Rabies G coding plasmid (pHMC122) ($2 \times 10^4$ FFU R-PV-GFP pseudovirion particles) or mock plasmid (pCDNA3.1), were used for infecting the target cells. Monolayers of Vero and HEK293T cells grown in 96-well plates were used as target cells to find the permissiveness to R-PV-GFP infection at a MOI of 1. Cells were imaged 24h post-infection.

G*ΔG-GFP rVSV used for initial transduction. Infection of the target cells with R-PV-GFP generated detectable cytoplasmic fluorescence within 4–6 h of infection. Fluorescence intensity became very bright by 12–18 h post-infection, when imaging was done. As observed, both Vero and HEK293T cells were permissive to infection with R-PV-GFP; however, HEK293T cells showed a better pseudovirion infection and further experiments were carried out using these cells.

## Susceptibility of R-PV-GFP to neutralization by specific antibodies

The ability of specific antibodies to neutralize R-PV-GFP was evaluated using anti-rabies G monoclonal antibody 17C7 as well immune serum from a vaccinated individual. As shown in Figs 2 and 3, the 17C7 antibody as well as the immune serum from vaccinated individual neutralized R-PV-GFP in a concentration dependent manner, with complete neutralization at lower dilutions. Negative control antibody 4G2 as well as the serum from individual who was not vaccinated with anti-rabies vaccine were unable to neutralize R-PV-GFP indicating the specificity.

## Generation and evaluation of R-PV-SEAP

SEAP expressing rabies pseudovirions (R-PV-SEAP) were generated as per the scheme shown in Fig 4A. The culture supernatant from the producer HEK293 cells transfected with pHMC122 and transduced with G*ΔG-SEAP rVSV were collected after 18h. It was subjected to $TCID_{50}$ evaluation as described in the methods section. The R-PV-SEAP virus stock produced had a $TCID_{50}$ ranging from 5.1 to $6.3 \times 10^4$ $TCID_{50}$ per ml.

Neutralizing antibodies detect the functional conformation of the rabies glycoprotein G over the viral envelope. The incorporation of rabies G protein trimers on the rVSV was assessed using immunostaining. R-PV-SEAP pseudovirion bound to HEK293T cells was detected using the neutralizing antibody 17C7 with antihuman Alexa Fluor 488 antibody as the secondary antibody. Cells stained after 30 min infection showed distinct fluorescent speckles that co-localize with the cell membrane. This confirmed the presence of rabies glycoprotein trimers on the surface of R-PV-SEAP pseudovirions (Fig 4B).

Varying concentration of the R-PV-SEAP was used to infect the HEK293T target cells and the SEAP activity in the supernatant was assayed. As shown in Fig 4C, there was a dose-dependent increase in SEAP activity indicating infection by the R-PV-SEAP pseudovirions. Further, the optimum concentration of the R-PV-SEAP pseudovirions for antibody neutralization assays were determined by varying the virus concentration from 25 to 2000 $TCID_{50}$/ well and using a fixed dilution (1:64) of a positive control antibody (17C7; 2IU/ml) and a negative control antibody (4G2). As shown in Fig 4D, there was complete neutralization by the specific antibody in all concentrations of R-PV-SEAP tested as indicated by the significant decrease in the alkaline phosphatase activity. Whereas, in the negative control antibody treated viruses, the SEAP activity was retained indicating no neutralization. Also, the OD values increased proportional to the concentration of the virus used, indicating the dose response to infection. It was observed that use of 100 $TCID_{50}$ R-PV-SEAP resulted in a moderate SEAP activity falling in the median range of absorbance; and increasing the virus concentration

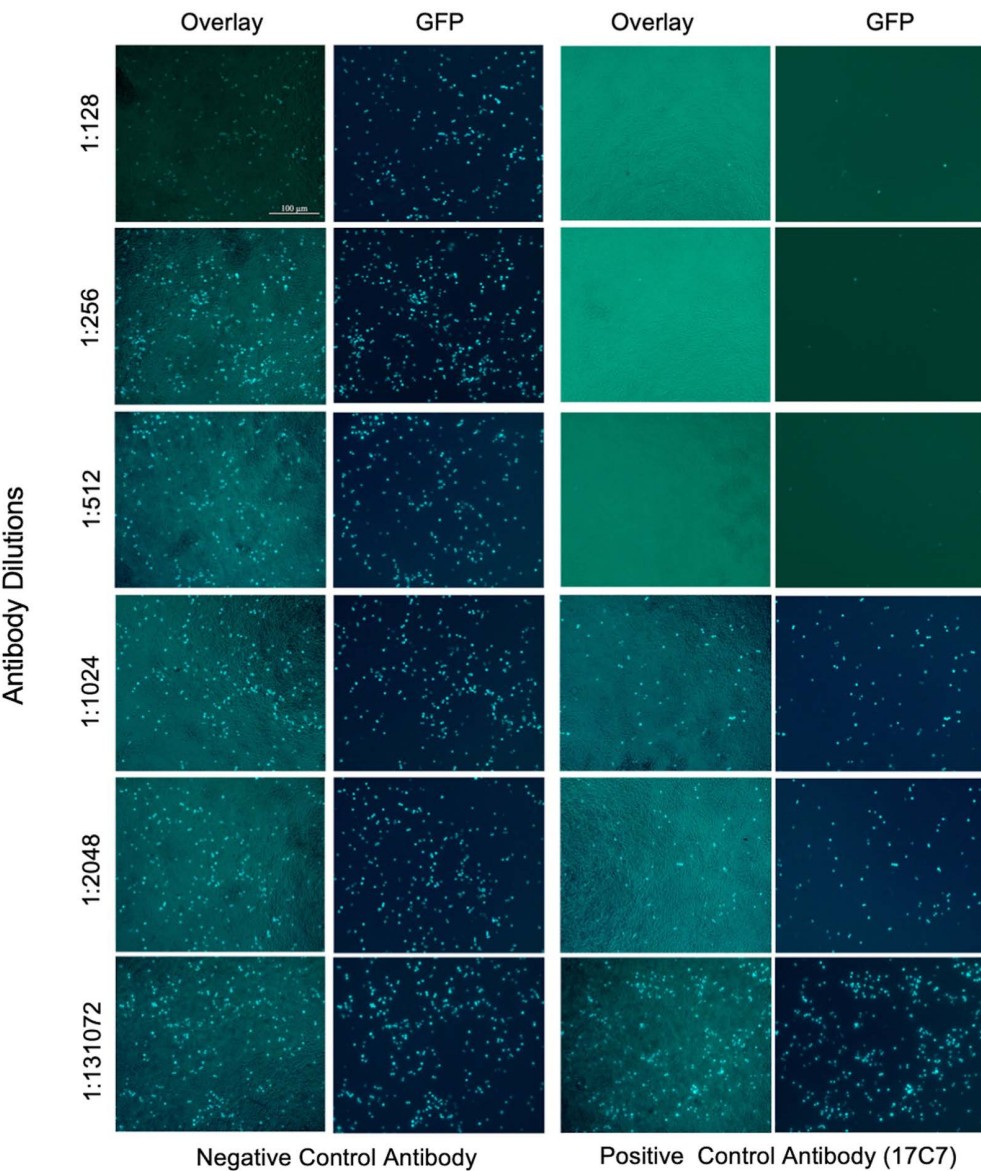

**Fig 2. Susceptibility of R-PV-GFP to neutralization by anti-rabies G monoclonal antibody.** Indicated two-fold dilutions of the anti-Rabies monoclonal antibody 17C7 were preincubated with the R-PV-GFP pseudovirions ($2 \times 10^4$ FFU) for 1hr at 37 °C; and were used to infect HEK293T target cells. Similar dilutions of a control antibody (4G2) was used as the negative control. Live cells were imaged 12 h post-infection using Zeiss inverted fluorescence microscope equipped with FITC excitation-emission filter sets.

beyond this resulted in saturation of the values. Hence, we decided to use this concentration as the optimal dose of the virus for subsequent serum neutralization assays.

## Susceptibility of R-PV-SEAP to neutralization by reference antibodies and human serum samples with varying RFFIT titre

Like R-PV-GFP, R-PV-SEAP pseudovirions also were effectively neutralized by varying dilutions of the reference antibody 17C7. We observed that the SEAP-based neutralization assay had a dynamic range between $OD_{405}$ 0.10 to 2.00 with the use of $100TCID_{50}$ of the R-PV-SEAP pseudovirions; and

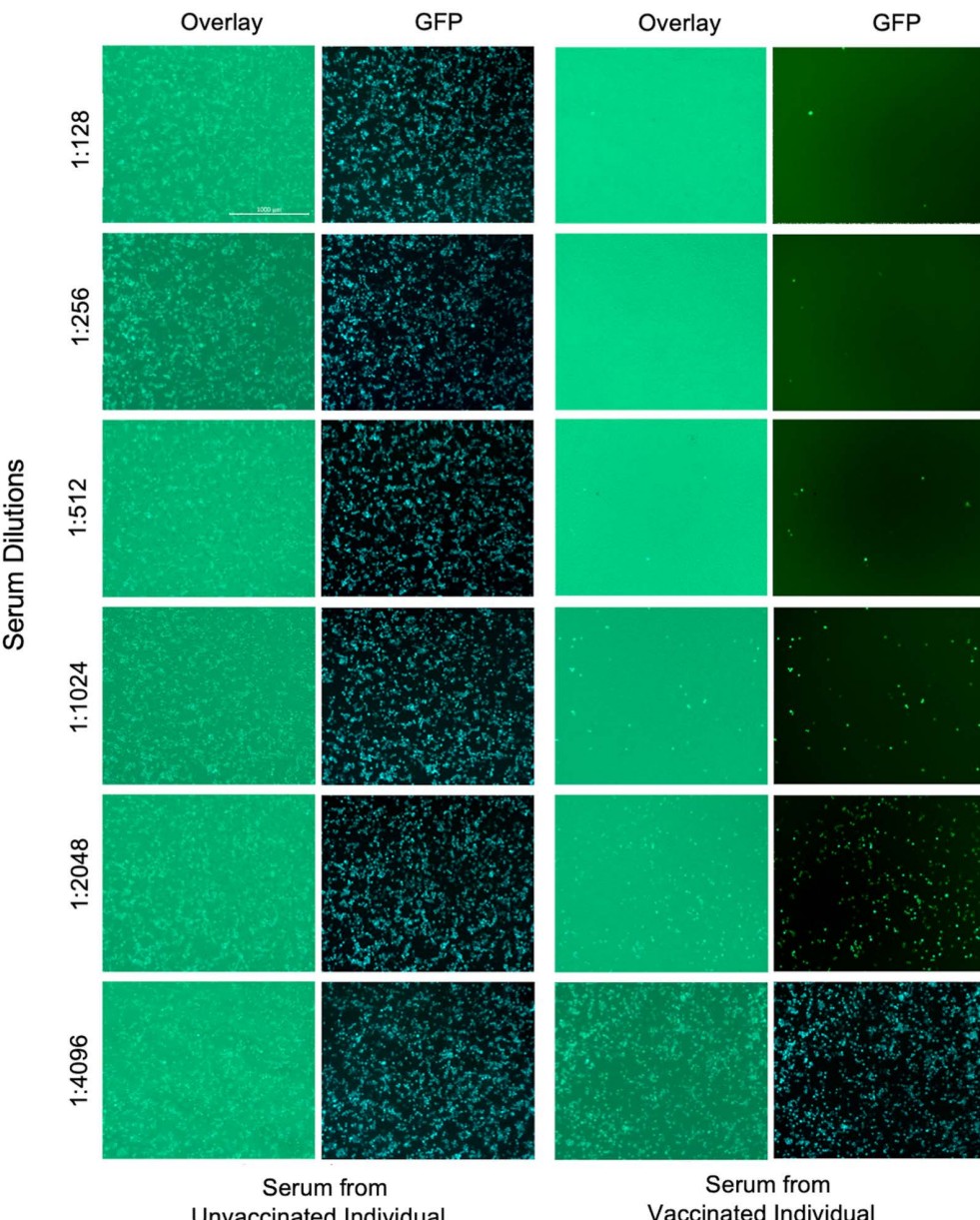

**Fig 3. Susceptibility of R-PV-GFP to neutralization by immune serum from vaccine recipient.** Indicated two-fold dilutions of the serum from an individual who received post-exposure prophylactic vaccination in 100 µl of 2% FBS containing DMEM were pre-incubated with an equal volume of the medium containing $2 \times 10^4$ R-PV-GFP pseudovirion particles for 1hr at 37 °C; and were used to infect monolayers of HEK293T target cells cultured in 96-well plates. Similar dilutions of serum from an unvaccinated individual was used as a negative control. Live cells were imaged 12 h post-infection using Zeiss inverted fluorescence microscope equipped with FITC excitation-emission filter sets.

any value below $OD_{405}$ 0.20 indicates 95% neutralization. As shown in the Fig 5A and 5B, lower dilutions of 17C7 and WHO reference serum samples showed complete neutralization, and OD values were close to cell control (OD-0.10). An increase in the dilution also reflected an increase in the SEAP activity proportional to the non-neutralized pseudovirions infecting the target cells.

 Before applying the assay to a larger set of samples, we evaluated the performance of the R-PV-SEAP PVNT with ten human serum samples with RFFIT values arbitrarily set as high

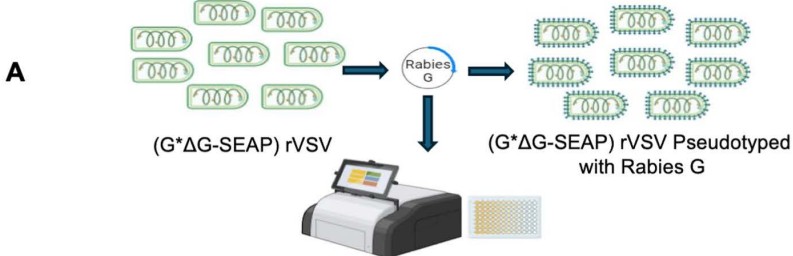

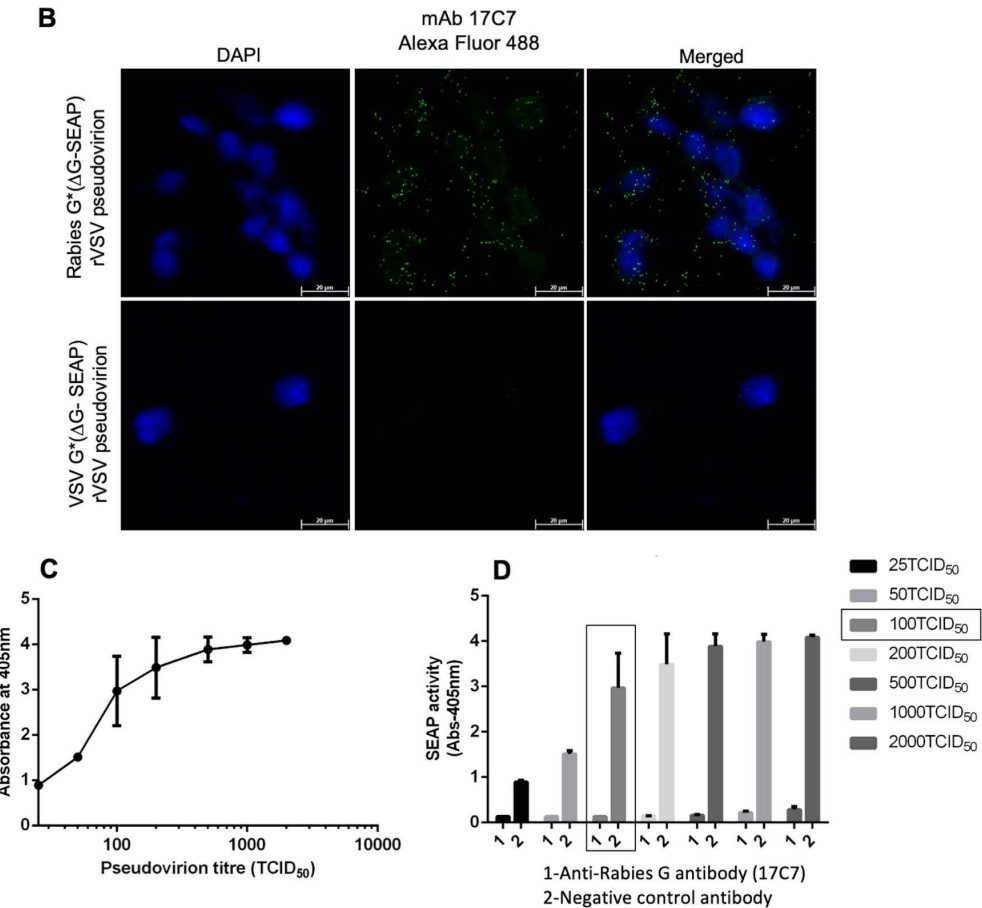

**Fig 4. Generation and evaluation of R-PV-SEAP. A**. Scheme showing generation of Rabies pseudovirus expressing secreted alkaline phosphatase as reporter (R-PV-SEAP) using G*ΔG-SEAP rVSV and full-length Rabies virus G protein expressing plasmid pHMC122 (Generated using Biorender; https://www.biorender.com/). **B**. Immunofluorescence analysis of R-PV-SEAP infection on HEK 293T cells. The panel represents the infection of target cells with 2500 $TCID_{50}$ Rabies G*(ΔG- SEAP) rVSV pseudovirion (top panel) and VSV G*(ΔG- SEAP) rVSV pseudovirion (bottom panel) respectively. The infected cells were fixed for immunofluorescence with 17C7 mAb (primary antibody) followed by anti- human Alexa fluor 488 secondary antibody. Pseudoviruses on the surface and within the cells are stained green and nuclei stained blue with DAPI. Scale bar, 20 μM. **C**. SEAP activity in the culture supernatants HEK293T target cells upon infection with increasing concentrations of R-PV-SEAP pseudovirions 24h post-infection. Each value is a mean ± SD of absorbance from two independent experiments (N = 2). **D**. Neutralization efficiency of anti-Rabies virus G monoclonal antibody (17C7) and a negative control antibody (4G2) at 1:64 dilution against varying concentrations of R-PV-SEAP pseudovirions. The antibodies were pre-incubated with the R-PV-SEAP pseudovirions for 1hr at 37 °C; and were used to infect HEK293T target cells. Cell culture supernatants were collected at 24h post-infection and subjected to SEAP assay. Each value is a mean ± SD of absorbance from two independent experiments (N = 2).

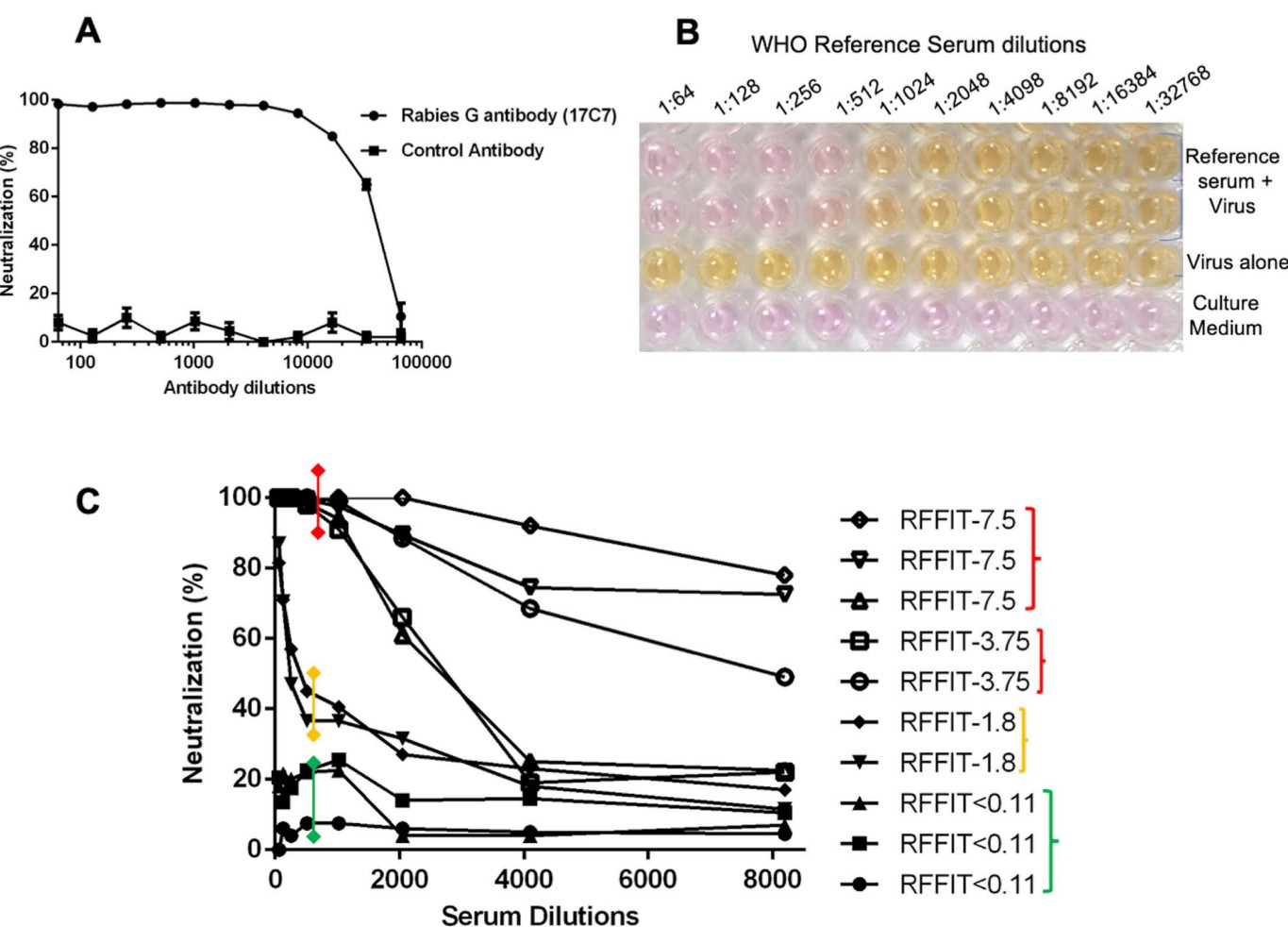

**Fig 5. Neutralization efficiency of R-PV-SEAP by reference antibodies and human serum samples with varying RFFIT titre. A**. $100 TCID_{50}$ of R-PV-SEAP pseudovirions were subjected to neutralization by varying dilutions of the 17C7 anti-Rabies G monoclonal antibody and a non-neutralizing negative control antibody (4G2) by incubating at 37 °C for 1hr; and then infecting HEK293T target cells. At 12h post-infection, the culture supernatants were subjected to SEAP assay and percentage neutralization was calculated against pseudovirus infected controls. Each value represents a mean ± SD of absorbance from three independent experiments. **B**. A representative colorimetric reaction plate. $100 TCID_{50}$ of the R-PV-SEAP pseudovirions were subjected to neutralization by incubating with varying dilution of a WHO reference serum (2IU/mL) for 1hr at 37 °C; and subsequently used to infect target HEK293T cells. Media controls and cells infected with un-neutralized pseudovirus were kept as controls. Visualization of the SEAP activity in the culture supernatants after 12h are shown. **C**. Neutralization efficiency of human serum samples from rabies vaccinated individuals with varying levels of Rabies Fluorescent Focus Inhibition Test (RFFIT) positivity against $100 TCID_{50}$ of R-PV-SEAP pseudovirions. Each data point represents mean ± SD of absorbance from two independent experiments.

(>7.5 IU/ml; 3 Nos.), low (3.75 IU/ ml; 2 Nos.), border-line (1.8-0.5 IU/ml; 2 Nos.) and negative ranges (< 0.5 IU per ml; 3 Nos). As shown in Fig 5C, the PVNT assay clearly distinguished the positive samples of the high and low RFFIT values with a positive neutralization whereas samples with borderline RFFIT values showed a variable neutralization when a cut off value of 100% was set. RFFIT negative samples were non-neutralizing in the R-PV-SEAP PVNT assay also.

## Concordance of Pseudovirus neutralization titres (PVNT) with RFFIT titres in a larger set of human serum samples

PVNT analyses were carried out in a set of 71 human samples diluted at 1:64 as for the RFFIT assay and the results were compared at two lower cut-offs, 50% and 75%. As indicated in the

Tables 1 and S1 and Fig 6A and 6B the number of samples showing positive neutralization were identical for samples with high neutralizing antibody titre, as indicated by the higher RFFIT values. However, in the case of samples with border-line RFFIT neutralization values (indicated by a rectangular box in Fig 6A and 6B), there were variability. While a few of the samples showed complete neutralization in PVNT, there were many samples that showed lower levels of neutralization. The 50% and 75% cut-off values affected only these samples with respect to their PVNT positivity.

**Table 1. Comparison of PVNT and RFFIT results of human serum samples at 1:64 serum dilution.**

|                    | RFFIT | PVNT-50% | PVNT-75% |
|--------------------|-------|----------|----------|
| Numbers Positive   | 61    | 60       | 59       |
| Numbers Negative   | 10    | 11       | 12       |

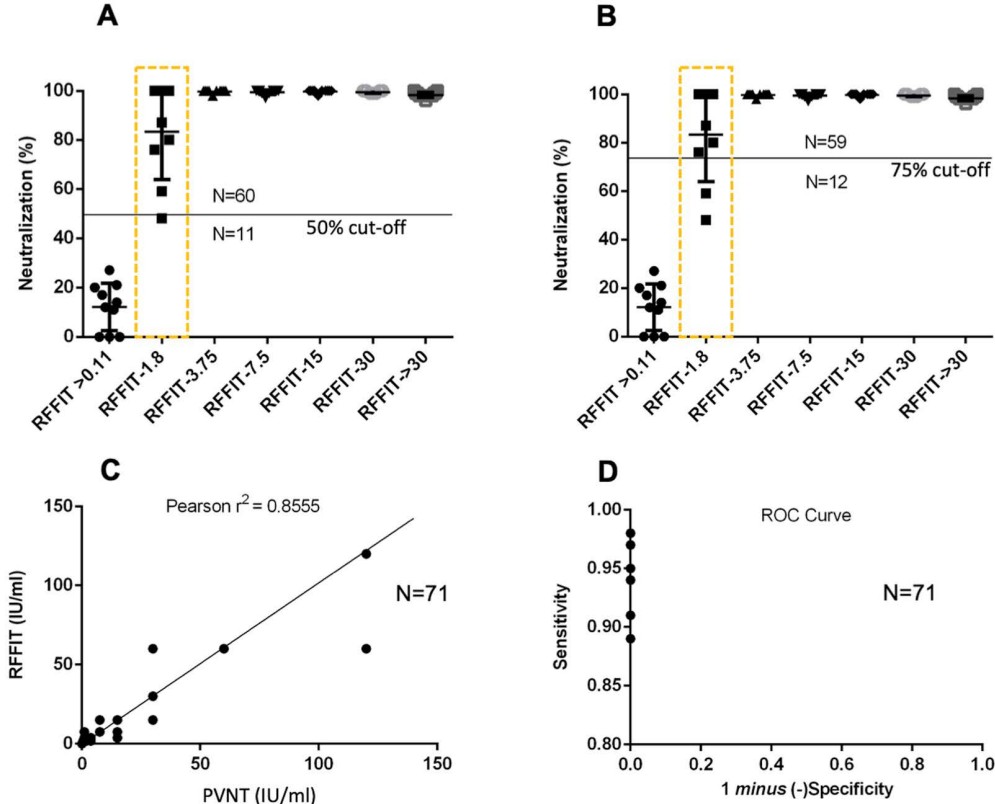

**Fig 6. Concordance of Pseudovirus neutralization titres (PVNT) with RFFIT titers.** Neutralization efficiency of human serum samples from 71 rabies vaccine recipients with known RFFIT titres were evaluated against 100 $TCID_{50}$ of R-PV-SEAP and plotted. **A**. Number of samples showing positivity at 50% neutralization efficiency cut-off. Border-line positive samples are indicated by the rectangular box. **B**. Number of samples showing positivity at 75% neutralization efficiency cut-off. Individual values calculated as mean from reading in duplicates are represented as data points. **C**. Correlation of pseudovirus neutralization titres (PVNT) and RFFIT titres of 71 human serum samples. PVNT values were converted to International Units with respect to the WHO reference serum neutralization values and plotted. A few of the data points represent multiple samples having identical neutralization titers. **D**. Receiver operator characteristics (ROC) of the PVNT assay. Sensitivity and specificity values of the PVNT Assay using the 71 samples at varying positivity cut-offs were calculated and plotted (Table 2). A few of the data points represent multiple samples having identical values.

RFFIT and PVNT neutralization titres calculated with respect to WHO reference serum samples were used for the Pearson's Correlation analysis. The assays has a correlation of 0.855 as shown in Fig 6C. The assay showed consistent characteristics such as specificity and sensitivity (Tables 1 and 2) in a wide range of RFFIT-validated samples. Analysis of the Receiver Operator Characteristics (ROC) (Fig 6D and Table 2) indicated that the assay had 100% specificity at any neutralization cut-off settings, but varying levels of sensitivity. As shown in Table 2, sensitivity varied from 0.89 to 0.98 with decreasing neutralizing cut-offs from 90% to 50%. The negative predictive value of the assay was lower (0.59 to 0.91) and increased at lower cut-offs. However, it had 100% positive predictive value at any cut off values above 50%, which can be used in the evaluation of unknown samples by this PVNT.

## Discussion

WHO strives to achieve zero dog-mediated rabies death globally by 2030 as envisaged by the UN Sustainable Development goals [27,28]. It strategizes to focus on mass vaccination of dogs, better wound management and improved access to effective post-exposure prophylaxis. This, in turn, has generated momentum for better vaccine coverage in humans and animals; and a need to ensure protective seroconversion in vaccine recipients.

Virus neutralization assays, such as RFFIT and FAVN, are essential for measuring neutralizing antibodies against the rabies virus in both humans and animals. They are widely used to evaluate new vaccines and vaccination schedules, confirm adequate seroconversion in immunocompromised individuals following vaccination, and determine booster requirements for individuals at high risk of rabies exposure. In human cases, these assays also support laboratory confirmation of clinically suspected rabies. For animals, they are recommended to meet international movement and trade regulations, verify immunity in vaccinated pets and livestock, support wildlife rabies control programs, and facilitate research on cross-species vaccine efficacy [4].

In the present study we aimed to generate an alternate assay that avoids the use of live rabies virus, the major limiting factor in the wide-spread use of RFFIT-based neutralization assays. Pseudoviruses are advantageous to live virus in many aspects including safety and biocontainment. They are much safer to use in a laboratory setting due to their restricted infectivity and lack of virulent viral components [17–19]. Even though alternate assays were reported previously using lentivirus- based pseudoviruses [21,22], the low titre of the pseudoviruses produced and the difficulty in adopting this to large-scale screening were major impediments. In our study, we overcame the first issue with the use of a rVSV-based system [23] that necessitates only transfection of single plasmid expressing the rabies glycoprotein G in the HEK293 producer cells. The conventional pseudovirion generation techniques involves the use of multiple plasmids and transfections to achieve all the components needed for the pseudovirion assembly. Since the incorporation of these multiple plasmids into the same cell during transfection remains a rare event, the process of viral assembly is very inefficient and results in lower and inconsistent viral titres. We incorporated the high throughput adaptability

**Table 2.** PVNT Assay performance of RFFIT validated human serum samples (n = 71) at different neutralization cut-offs.

| Neutralization (%) | >90 | 85–89 | 80–84 | 75–79 | 70–74 | 65–69 | 60–64 | 55–59 | 50–54 |
|---|---|---|---|---|---|---|---|---|---|
| Sensitivity | 0.89 | 0.89 | 0.91 | 0.94 | 0.95 | 0.95 | 0.97 | 0.97 | 0.98 |
| Specificity | 1 | 1 | 1 | 1 | 1 | 1 | 1 | 1 | 1 |
| Positive Predictive Value (PPV) | 1 | 1 | 1 | 1 | 1 | 1 | 1 | 1 | 1 |
| Negative Predictive Value (NPV) | 0.59 | 0.59 | 0.62 | 0.71 | 0.77 | 0.77 | 0.83 | 0.83 | 0.91 |

into the system by using a rVSV that expresses a quantifiable reporter (SEAP) upon infecting the target cells which can be measured in conventional ELISA readers in 96-well format.

Our initial analysis with rVSV encoding a GFP-based reporter indicated production of a high titre rabies pseudovirus (with a fluorescent focus unit, FFU of 1.2- $2.0 \times 10^5$/ml) that was amenable to neutralization by specific antibodies (Figs 1–3). Subsequently, using the rVSV with SEAP reporter (R-PV-SEAP), we could reproduce the high titre pseudovirus production in different batches, with viral titres ranging from 5.1 to $6.3 \times 10^4$ TCID$_{50}$ per ml that gave a consistent SEAP absorbance value. This is being enhanced further by generating HEK293 producer cells stably expressing rabies G protein through selection, which will eliminate the need for recurrent transfection while producing the R-PV-SEAP. In our experiments, we could also demonstrate the efficient binding of the pseudovirus produced on to the cell membrane of the HEK293T target cells by visualizing the immunostained virus particles in confocal microscopy (Fig 4B).

We observed that 100 TCID$_{50}$ is an optimal titre for the PVNT assays (Fig 4D), though the R-PV-SEAP could be produced in higher titres. In neutralization assays with reference antibodies and human serum samples, the results were reproducible with a consistent dose-response to different antibody dilutions (Fig 5A and 5B). PVNT assays using a sub-set of serum samples with varying RFFIT titres could clearly distinguish positive and negative samples but with variable results in the borderline samples (Fig 5C). Overall, the PVNT assay showed promising results in the initial analysis.

We extended the study further with 61 RFFIT positive human serum samples and 10 negative samples which were subjected to the PVNT analysis (Fig 6A and 6B). We could reproduce the results observed in the pilot study with a subset of samples. The border-line RFFIT positive samples showed a variable response with many showing complete neutralization and some of them showing a low neutralization. This variability has also affected the calculated sensitivity of the assay as well as the Pearson's correlation with the RFFIT titres (Fig 6C). However, the encouraging observation was that the PVNT assay had 100% specificity and positive predictive values in all the cut-offs set, as indicated in the ROC analysis (Table 2 and Fig 6D). Any assay with false positives cannot be recommended for evaluating protective immune response against a fatal disease like rabies. Hence, this strength of PVNT makes it a robust alternative to RFFIT in spite of a lower sensitivity.

The study has some limitations. One of them was that in the ROC analysis, we could not carry out an area-under curve (AUC) estimation as all the data points were confined to the Y-axis since all the samples had 100% specificity. Secondly, we could include only ten RFFIT-validated negative samples in the assay which might be responsible for a reduced estimation of the sensitivity of the assay. The total sample size was also limited to 71. Further studies are being undertaken to address these issues.

An overall scheme of the assay developed in this study is indicated in Fig 7. This rVSV-based rabies PVNT assay with SEAP as the detection system described here is comparable to the live virus-based RFFIT assay. It overcomes major limitations of the RFFIT assay [29], most importantly the need to use live rabies virus. It eliminates the need for a much costly fluorescent microscope, and the need for fluorescent-based staining and cumbersome manual examination of the images under a microscope. Also, in RFFIT assays, additional care needs to be taken to avoid bleaching and quenching of the fluorescence which also limit the repeatable reading of the results. Another major advantage of the PVNT assay we observed was the significantly reduced turnaround time. In less experienced settings, RFFIT assays may take 48-72h before the results are read [13]. In PVNT, the SEAP expression can be detected by 12h post-neutralization. Further, it has a high-throughput adaptability as it can be done in multi-well plates and can be read using commercial plate readers for measuring absorbance.

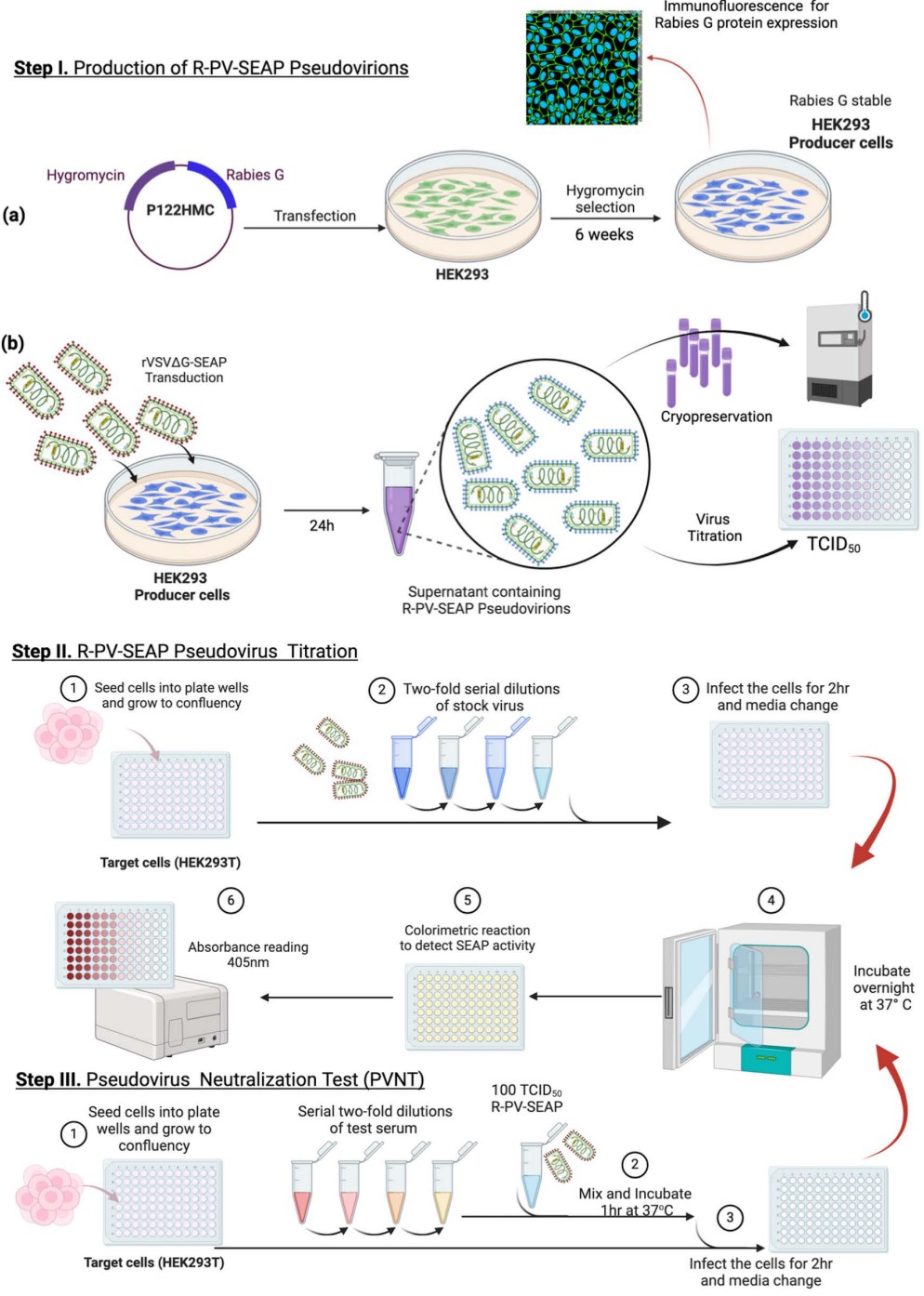

**Fig 7. Overall Scheme of R-PV-SEAP pseudovirion production and pseudovirus neutralization test (PVNT).** (Generated using Biorender; https://www.biorender.com/).

This also eliminates the subjectivity linked with imaging and ensures reproducibility and easy documentation.

Selection of appropriate assays are of paramount importance in evaluation of protective immune response to vaccines against rabies [30]. Overall, our results show that this rabies

PVNT is a reproducible and highly specific assay with an objective and documentable readout that can serve as a suitable alternative to RFFIT to evaluate protective response to vaccination.

## Supporting information

**S1 Table. RFFIT and PVNT titres of human serum samples.**
(PDF)

**S1 Data. Raw Data for graphs.**
(DOCX)

## Acknowledgments

The authors acknowledge the excellent technical support by Mr. Gopikrishnan K in acquiring confocal microscopic images. The authors acknowledge Dr. Erica Ollman Saphire, La Jolla Institute of Immunology, USA for sharing key reagents for the study.

## Author contributions

**Conceptualization:** Easwaran Sreekumar.

**Formal analysis:** Santhik S Lupitha, Geetu Rose Varghese, Lekshmi J Das, Ashwini M Ananda.

**Funding acquisition:** Easwaran Sreekumar.

**Investigation:** Santhik S Lupitha, Geetu Rose Varghese, Lekshmi J Das, Priya Prabhakaran, Ashwini M Ananda.

**Methodology:** Reeta S Mani, Easwaran Sreekumar.

**Project administration:** Reeta S Mani, Easwaran Sreekumar.

**Resources:** Reeta S Mani, Easwaran Sreekumar.

**Supervision:** Reeta S Mani, Easwaran Sreekumar.

**Validation:** Reeta S Mani, Easwaran Sreekumar.

**Visualization:** Santhik S Lupitha, Geetu Rose Varghese.

**Writing – original draft:** Santhik S Lupitha.

**Writing – review & editing:** Reeta S Mani, Easwaran Sreekumar.

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
