## [Decision Letter · Decision Letter 0]

28 Jan 2025

Response to Reviewers
Revised Manuscript with Track Changes
Manuscript

Shaden Kamhawi

co-Editor-in-Chief

Paul Brindley

co-Editor-in-Chief

**Journal Requirements:**

At this stage, the following Authors/Authors require contributions: Santhik S Lupitha, Geetu Rose Varghese, Lekshmi J Das, Priya Prabhakaran, Ashwini M Ananda, Reeta S Mani, and Easwaran Sreekumar. Please ensure that the full contributions of each author are acknowledged in the "Add/Edit/Remove Authors" section of our submission form.

2) Please ensure that the Title in your manuscript file and the Title provided in your online submission form are the same.

- ® on page: 7.

Potential Copyright Issues:

- Figures 1A and 4A; Please confirm whether you drew the images / clip-art within the figure panels by hand. If you did not draw the images, please provide a link to the source of the images or icons and their license / terms of use; or written permission from the copyright holder to publish the images or icons under our CC BY 4.0 license. Alternatively, you may replace the images with open source alternatives. See these open source resources you may use to replace images / clip-art:

**Reviewers' comments:**

**Key Review Criteria Required for Acceptance?**

**Methods**

-Are the objectives of the study clearly articulated with a clear testable hypothesis stated?

-Is the study design appropriate to address the stated objectives?

-Is the population clearly described and appropriate for the hypothesis being tested?

-Is the sample size sufficient to ensure adequate power to address the hypothesis being tested?

-Were correct statistical analysis used to support conclusions?

-Are there concerns about ethical or regulatory requirements being met?

Reviewer #1: The methods were clearly stated to accomplish the stated goal of development of a valid rabies pseudotype neutralization assay. As weel the design was completed, including both detection in cells and supernatant. There are minor points/details and would allow improved description of the experiments:

Under Titration of R-PV-SEAp pseudovirus stocks, what serial dilutions of the stocks were used?

In the ethics statement, was SRIG was used in the RFFIT test for the human serum samples tested?

Which WHO reference serum was used in the PVNT assay, was it the same WHO reference serum used to qualify the 17C7 monoclonal?

In the paragraph where the PVNT is first described, it mentions that the samples were diluted in 2% FBS containing DMEM but there is no mention of what diliution was used. In 2 paragraphs later a 2-fold serial dilution was performed, referencing Smith, 1973 (the basis of the RFFIT method in the WHO manual) states a 5-fold serial dilution is used. Suggest, referencing the modified method (Ashwath) only.

What was the cell count used for the HEK293T cells?

Reviewer #2: The article “Assessing anti-rabies vaccine response in humans: A rapid and high-throughput— A rapid and high-throughput-adaptable neutralization assay as an alternative to Rabies Fluorescence Focus Inhibition test (RFFIT)” describes a more practical alternative to RFFIT that avoids using live rabies virus, a fluorescence microscope, and uses SEAP expressed by the pseudovirus as a reporter.

Dear authors, congratulations on the submitted manuscript. It is well presented, and the work represents a very interesting alternative system to measure rabies-neutralizing antibodies.

The objectives and study design it is appropriate.

When reading, I was expecting also a comparison with the R-PV-GFP virus.

**Results**

-Does the analysis presented match the analysis plan?

-Are the results clearly and completely presented?

-Are the figures (Tables, Images) of sufficient quality for clarity?

Reviewer #1: The result section described the outcome of the generation of the pseudotypes and the ability of the pseudotypes to measure neutralization by previously determined positive RVNA samples. The figures are effective in displaying the results. Under the section Susceptibility of R-PV-SEAP to neutralization by reference.... the cut-offs used for the assay are not discussed, It would be helpful also if descriptions of high, low, borderline and negative RFFIT ranges were given. At the bottom o fpage 11 the sentence: The assay showed con characteristics such as..., what is meant by con?

Reviewer #2: When reading, I was expecting also a comparison with the R-PV-GFP virus.

Please find some small suggestions below, based on the pdf version received:

Page 1)

Title: Please check that two words are together, at least in the loaded file.

Page 7)

In “Titration of R-PV-SEAP," when mentioning “serial dilution," which dilution are you referring to? Twofold or tenfold dilution?

As far as I understand, the experiments are done in preformed cell monolayers; is correct?

Page 10)

Please check whether R-PV-SEAP should replace RPV-SEV.

Page 13)

When you mention that the titer reached with the R-PV-GFP is produced in high titer, are the same numbers as the R-PV-SEAP? Could you add this number to the discussion?

At which scale is it enough to prepare a viral stock for an X quantity of rabies-neutralizing antibody determinations? Just a comment, if possible.

Page 18)

In Figure 1, or the manuscript, the MOI (or infection dilution used) is not mentioned. Figure 1.A) Consider if it could be improved for those unfamiliar with the system.

Page 21)

In Figure 3, what is the titer of the virus mixed with the serum?

**Conclusions**

-Are the conclusions supported by the data presented?

-Are the limitations of analysis clearly described?

-Do the authors discuss how these data can be helpful to advance our understanding of the topic under study?

-Is public health relevance addressed?

Reviewer #1: The discussion method is written very well, and effectively expresses the reason for the study and why use of a rVSV-based system for obtaining high titre is preferred. The practical advantages were well covered. The importance of a rabies antibody assay having high specificity in spite of low sensitivity was adequately made. Perhaps the "equal or better" statement was overstated. Some of the downsides to the traditional RFFIT, while experience in some hands, does not apply to all laboratories that can report values within 24 to 48 hours, and not experience statin bleaching with new efficiency since the 1970s. Indeed, SNs can and are be read by instrumentation (Wu et al. JVirolMeth 2023 for example) --a more evened argument would be more factual. However, the main advantage is elimination of need for high containment was well covered in light of the zero by 30 effort.

Reviewer #2: Page 14) Besides the limitations you mentioned about the current state of this work, does R-PV-GFP work similarly to the R-PV-SEAP? Immunostaining should not be necessary because it expresses the GFP reporter well.

If you have information or data (using R-PV-GFP) similar to the one generated with the R-PV-SEAP, it could be of interest to those who are already doing RFFIT and have access to a fluorescence microscope.

**Editorial and Data Presentation Modifications?**

Reviewer #1: Minor comments:

In the Abstract, correct the name of RFFIT, it is Rapid Fluorescent Focus Inhibition test (see WHO Manual).

Clarify the sentence in the Abstract: In samples with border-line RFFIT values (what are borderline?), concordance was low, nevertheless, the results were tending to be negative (waht results, borderline RFFIT or PVNT results?).

Using the term protection in terms of rabies antibody level should be defined. There is no known/proven level of rabies antibody, the 0.5 IU/mL was assigned by review of response to vaccination and represents a robust level.

In the Introductions the last sentence reads oddly. Perhaps what is meant is RFFIT-validated rabies antibody levels in human serum samples.....

Reviewer #2: (No Response)

**Summary and General Comments**

Reviewer #1: Well-written, worth study to be published. Expected to be useful to the zero by 30 effort and generation of useful data for future studies and policies.

Reviewer #2: (No Response)

PLOS authors have the option to publish the peer review history of their article (what does this mean? ). If published, this will include your full peer review and any attached files.

**Do you want your identity to be public for this peer review?** For information about this choice, including consent withdrawal, please see our Privacy Policy .

Reviewer #1: No

Reviewer #2: **Yes: ** Carlos Adolfo Palacios

**Figure resubmission:****Reproducibility:** To enhance the reproducibility of your results, we recommend that authors of applicable studies deposit laboratory protocols in protocols.io, where a protocol can be assigned its own identifier (DOI) such that it can be cited independently in the future. Additionally, PLOS ONE offers an option to publish peer-reviewed clinical study protocols. Read more information on sharing protocols at https://plos.org/protocols?utm_medium=editorial-email&utm_source=authorletters&utm_campaign=protocols

---

## [Decision Letter · Decision Letter 1]

21 Mar 2025

Dear Dr. Sreekumar,

We are pleased to inform you that your manuscript 'Assessing anti-rabies vaccine response in humans: A rapid and high-throughput adaptable, pseudovirus-based neutralization assay as an alternative to Rapid Fluorescence Focus Inhibition test (RFFIT)' has been provisionally accepted for publication in PLOS Neglected Tropical Diseases.

Best regards,

Sergio Recuenco, DrPH

Academic Editor

Andrea Marzi

Section Editor

Shaden Kamhawi

co-Editor-in-Chief

Paul Brindley

co-Editor-in-Chief

Reviewer's Responses to Questions

**Key Review Criteria Required for Acceptance?**

**Methods**

-Are the objectives of the study clearly articulated with a clear testable hypothesis stated?

-Is the study design appropriate to address the stated objectives?

-Is the population clearly described and appropriate for the hypothesis being tested?

-Is the sample size sufficient to ensure adequate power to address the hypothesis being tested?

-Were correct statistical analysis used to support conclusions?

-Are there concerns about ethical or regulatory requirements being met?

Reviewer #2: The described assay provides a significantly faster turnaround time of under 24 hours, making it ideal for automation in large-scale screening efforts. Importantly, it removes the necessity of handling live rabies virus, a critical advantage when evaluating protection levels in vaccinated individuals.

**Results**

-Does the analysis presented match the analysis plan?

-Are the results clearly and completely presented?

-Are the figures (Tables, Images) of sufficient quality for clarity?

Reviewer #2: The presented results aim to eliminate the need for both the rabies virus and a fluorescence microscope, offering a practical solution for environments lacking the facilities required to maintain the necessary biosafety level.

The results are well presented.

**Conclusions**

-Are the conclusions supported by the data presented?

-Are the limitations of analysis clearly described?

-Do the authors discuss how these data can be helpful to advance our understanding of the topic under study?

-Is public health relevance addressed?

Reviewer #2: The authors' conclusions are accurate, well presented, and it is anticipated that this approach could be widely adopted by laboratories to produce additional results and test the methodology across various research/diagnostics settings within this field.

**Editorial and Data Presentation Modifications?**

Reviewer #2: The authors have incorporated the suggested revisions, and the resulting improvements to the manuscript render it suitable for publication.

**Summary and General Comments**

Reviewer #2: (No Response)

PLOS authors have the option to publish the peer review history of their article (what does this mean? ). If published, this will include your full peer review and any attached files.

**Do you want your identity to be public for this peer review?** For information about this choice, including consent withdrawal, please see our Privacy Policy .

Reviewer #2: No

---

## [Editor Report · Acceptance letter]

Dear Dr. Sreekumar,

We are delighted to inform you that your manuscript, "Assessing anti-rabies vaccine response in humans: A rapid and high-throughput adaptable, pseudovirus-based neutralization assay as an alternative to Rapid Fluorescence Focus Inhibition test (RFFIT)," has been formally accepted for publication in PLOS Neglected Tropical Diseases.

Best regards,

Shaden Kamhawi

co-Editor-in-Chief

Paul Brindley

co-Editor-in-Chief
